# LLM Bandit: Cost-Efficient LLM Generation via Preference-Conditioned Dynamic Routing

## Abstract

The rapid advancement in large language models (LLMs) has brought forth a diverse range of models with varying capabilities that excel in different tasks and domains. However, selecting the optimal LLM for user queries often involves a challenging trade-off between accuracy and cost, a problem exacerbated by the diverse demands of individual queries. In this work, we present a novel framework that formulates the LLM selection process as a multi-armed bandit problem, enabling dynamic and intelligent routing of queries to the most appropriate model. Our approach incorporates a preference-conditioned dynamic routing mechanism, allowing users to specify their preferences at inference time, thereby offering a customizable balance between performance and cost. Additionally, our selection policy is designed to generalize to unseen LLMs, ensuring adaptability to new models as they emerge. Experimental results demonstrate that our method achieves significant improvements in both accuracy and cost-effectiveness across various LLM platforms, showcasing the potential of our framework to adaptively optimize LLM selection in real-world scenarios.

## 1 Introduction

By scaling up parameters and pretraining data, large language models (LLMs) have shown remarkable abilities across a wide range of tasks. However, while larger models tend to be more capable and versatile, they also come with higher costs — whether in terms of computational resources, API service fees, or increased response latency. Additionally, the growing number of domain-specific models often outperform these large models within their specialized areas. As a result, selecting the most suitable LLM for specific applications has become increasingly challenging for users, especially when faced with cost constraints.

Existing approaches to address the performance-cost dilemma typically fall into three categories. Firstly, ensemble methods (Jiang et al., 2023; Wang et al., 2023) combine responses from multiple LLMs to enhance overall performance. However, these methods require invoking multiple LLMs for each query, leading to substantially higher costs. Secondly, cascading approaches (Ramírez et al., 2024; Chen et al., 2023; 2020) sequentially invoke LLMs, starting with the least expensive and progressing to more costly models only if the initial responses are unsatisfactory. Lastly, routing mechanisms (Ding et al., 2024; Ong et al., 2024; Nguyen et al., 2024) direct user query to the most appropriate LLM without invoking any of them, offering a more cost-effective solution. In this work, we focus on the routing approach due to its flexibility and potential for optimizing cost-efficiency.

However, designing an effective routing module for real-world deployment presents several challenges. First, the routing mechanism must generalize effectively across diverse user queries, ensuring robust performance across various tasks and domains. Second, it should be capable of handling any set of models, as new LLMs are continually being developed, and users may wish to specify a particular set based on prior knowledge. Third, the routing mechanism needs to accommodate varying user preferences, as different users may prioritize cost, performance, or other factors differently depending on their specific requirements.

To address these challenges, we propose a preference-conditioned dynamic routing mechanism for LLMs. We frame the LLM routing task as a multi-armed bandit problem, where each user query is directed to the most suitable LLM based on a routing policy. Our policy leverages multi-objective

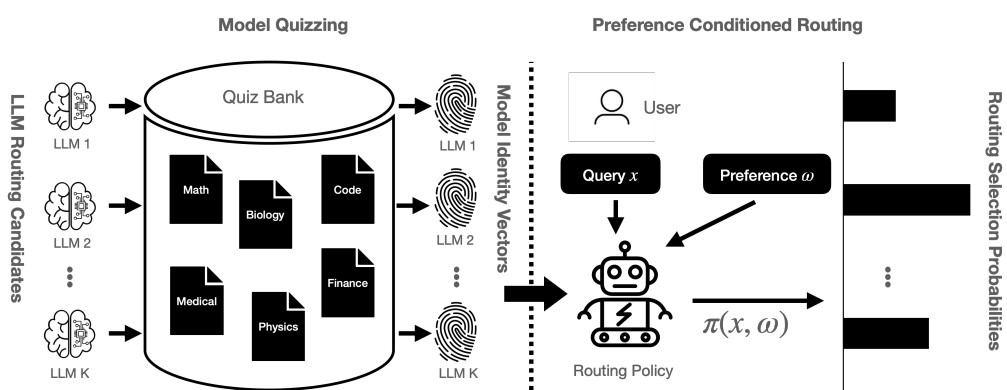

Figure 1: The overall framework of our proposed routing mechanism.

optimization, balancing performance and cost to suit individual preferences, such as budget constraints or quality requirements. By identifying the Pareto front of the optimization problem, our system provides users with an optimal trade-off between competing objectives. During deployment, users can specify their preferences, allowing the policy to dynamically adjust its routing decisions.

In addition, our routing policy is designed to be action-space aware, allowing it to generalize across any set of LLMs. This adaptability is crucial as it enables the policy to incorporate new models as they become available, adjusting its routing decisions without requiring extensive retraining or manual intervention. To facilitate learning of this action space-aware routing policy, we introduce a model identity vector representation that captures each model's capabilities across different tasks and domains. This identity vector is derived by predicting evaluation scores on a diverse set of prompts. To efficiently characterize new models, we further propose an efficient quizzing mechanism, which only requires evaluating on a small subset of selected prompts to generate accurate identity vectors.

Our contributions are summarized as follows: 1) We formulate the routing problem as a multi-objective optimization task, balancing the trade-off between performance and cost. 2) We propose a preference-conditioned routing mechanism that captures the Pareto front of the multi-objective optimization problem and adapts dynamically to user-specific preferences at inference time. 3) We develop an action-space aware routing policy capable of generalizing to arbitrary set of LLM models. 4) We leverage a wide range of existing evaluation data, including LLM evaluation leaderboards and pairwise model comparisons, to generalize the routing policy across a broad spectrum of user queries and LLM routing candidates. 5) We design an efficient quizzing mechanism to obtain model identity vectors for newly added models by evaluating only a small subset of prompts. 6) We demonstrate the effectiveness of our routing mechanism through comprehensive evaluation across multiple tasks and domains, showcasing superior performance.

## 2 METHOD

### 2.1 PROBLEM FORMULATION

Given a set of $K$ large language models (LLMs), denoted as $\{M_k\}_{k=1}^K$, we aim to develop a routing policy $\pi : \mathcal{X} \to \{1, \ldots, K\}$ that directs any query $x \in \mathcal{X}$ to the most appropriate LLM $M_k$. The routing decision results in a reward $s(x, k)$, reflecting the generation quality of $M_k$, while also incurring a cost $c_k$ for invoking the model. We formulate this task as a multi-objective optimization problem, where the reward vector is defined as $\mathbf{r}(x, k) = [s(x, k), -c_k]$. Our objective is to maximize the expected reward:

$$\pi^* = \arg\max_\pi \mathbf{J}_\pi = \arg\max_\pi \left[ \mathbb{E}_{x \sim p(x), k \sim \pi(x)} s(x, k), -\mathbb{E}_{x \sim p(x), k \sim \pi(x)} c_k \right]. \tag{1}$$

In the context of multi-objective optimization, where the objectives often conflict with each other, a single optimal policy that simultaneously maximizes all objectives does not exist. Instead, the optimal policy varies according to user preferences. For instance, a user with a limited budget might prioritize lower-cost models, even if this means compromising on generation quality. Conversely, a user who values high-quality outputs will have to accept higher costs. By introducing a preference

parameter $\boldsymbol{\omega}$, we define a scalarized reward $r_{\boldsymbol{\omega}}(x, k) = \boldsymbol{\omega}^T \mathbf{r}(x, k)$. The corresponding optimal policy $\pi_{\boldsymbol{\omega}}$ maximizes this scalarized expected reward $\mathbb{E}_{x \sim p(x), k \sim \pi_{\boldsymbol{\omega}}(x)} r_{\boldsymbol{\omega}}(x, k)$. The collection of all such optimal policies is referred to as the Pareto set, with their respective expected rewards $\mathbf{J}_{\pi_{\boldsymbol{\omega}}}$ forming the Pareto front.

## 2.2 Overall Framework

Our dynamic routing mechanism consists of two key components: a model quizzing component, which generates a model identity vector to capture each model's capabilities across various tasks and domains, and a preference-conditioned routing policy, which determines model selection probabilities based on user-specified preferences. Figure 1 provides an illustration of the framework. In the subsequent sections, we expound on the design details of these components.

## 2.3 Model Identity Vector

To accurately route queries to the appropriate LLM, we must first assess each LLM's strengths and weaknesses. For instance, queries requiring mathematical reasoning should be directed to LLMs with strong math capabilities, while financial queries should be routed to models fine-tuned for finance. We propose distilling each model's capabilities into a dense vector based on its performance across a diverse set of evaluation prompts.

Given a set of $N$ evaluation prompts, $X = \{x_n\}_{n=1}^N$, which span various domains and tasks, we collect the evaluation scores $Y_k = \{y_{kn}\}_{n=1}^N$ for each LLM $M_k$. The objective is to learn a model identity vector, $I_k \in \mathbb{R}^d$, that can effectively predict these evaluation scores. We employ a variant of the Item Response Theory (IRT) model (Hambleton & Swaminathan, 2013) combined with deep neural networks to perform this prediction. Unlike the traditional IRT model, we do not learn explicit representations for the prompts themselves; instead, we obtain prompt embeddings, $e_n$, using a pretrained text embedding model. These pretrained embeddings allow the IRT model to generalize effectively to unseen prompts, which, as we will discuss later, also enhances the generalization capability of the routing policy. Based on the prompt embedding $e_n$ and the model identity vector $I_k$, a deep neural network $f$ then predicts the evaluation scores as $f(e_n, I_k)$.

The training process closely follows that of traditional IRT, optimizing the binary cross-entropy loss. For non-binary evaluation scores, we binarize them by selecting a threshold that ensures the average performance across instances is comparable. Please refer to Appendix A.1.1 for details. Letting $\bar{y}_{kn}$ represent the binarized evaluation scores and $p_{kn} = \text{sigmoid}(f(e_n, I_k))$ represent the predicted probabilities, the loss function can be written as

$$\mathcal{L}_{irt} = \mathbb{E}_{x,k} \left[ -\bar{y}_{kn} \log p_{kn} - (1 - \bar{y}_{kn}) \log(1 - p_{kn}) \right]. \tag{2}$$

In addition to LLM evaluation benchmarks, where each model receives an evaluation score for a specific prompt, pairwise comparisons, where responses from two models to the same prompt are compared, are also widely used in the LLM evaluation literature. To increase prompt diversity, we propose leveraging these pairwise comparisons to aid in learning the model identity vectors. Given responses from two models, $M_{k1}$ and $M_{k2}$, and the annotations $z_n \in \{0, 1\}$, indicating the winner of the comparison, we employ a secondary neural network, $g$, to predict the winning probabilities via $p_n = \text{sigmoid}(g(e_n, I_{k1}) - g(e_n, I_{k2}))$. The associated loss function, similarly, follows a binary cross entropy formulation:

$$\mathcal{L}_{pair} = \mathbb{E}_{x,(k1,k2)} \left[ -z_n \log p_n - (1 - z_n) \log(1 - p_n) \right]. \tag{3}$$

The above loss function bears similarities to the reward model training loss, though our primary goal here is to learn the model identity vector rather than to directly optimize rewards. Note that we use the subscript $n$ to index prompts in both loss functions; however, these two types of evaluations typically occur on different prompts from separate datasets.

To enhance the generalizability of the model identity vectors to unseen models, we introduce a prior on the model embeddings $I_k$ using a variational inference formulation, treating the model identity vectors as latent variables. This approach adds an additional loss term, specifically the KL-divergence between the model identity vectors and their prior distributions:

$$\mathcal{L}_{KL} = \mathbb{E}_k \left[ D_{\text{KL}} \left( q(I_k) \| p(I_k) \right) \right]. \tag{4}$$

In practice, both the prior $p(I_k)$ and posterior $q(I_k)$ are modeled as Gaussian, with the mean and variance of the posterior being learnable parameters. Please see Appendix A.1 for further details.

## 2.4 Preference Conditioned Routing Policy

Recall that given a prompt $x_n$, a set of LLMs $\{M_k\}_{k=1}^K$, and a preference vector $\boldsymbol{\omega}$, the oracle routing policy $\pi^*$ would maximize the scalarized reward $r_{\boldsymbol{\omega}}(x, k) = \boldsymbol{\omega}^T \mathbf{r}(x_n, k) = \boldsymbol{\omega}^T [s(x_n, k), -c_k]$. However, the score $s(x_n, k)$ is typically not available, as it requires running inference for model $M_k$ on prompt $x_n$ and evaluating the response. A potential alternative is to estimate the scores without actually performing inference. For instance, we could use $f(e_n, I_k)$ as an estimate of the evaluation scores. This approach aligns conceptually with the methods in (Shnitzer et al., 2023; Hari & Thomson, 2023; Šakota et al., 2024). However, predicted scores can often be inaccurate, as the prediction model may fail to capture the finer nuances of model performance on specific prompts. Moreover, relying solely on predicted scores ignores the uncertainty in the prediction model itself. The discrepancy between estimated and actual utility can result in suboptimal routing decisions. To mitigate the impact of inaccurate predictions, Shnitzer et al. (2023) introduced a sophisticated meta-learning approach to reduce bias in the predicted scores.

Instead of relying solely on predicted scores, we advocate for learning a routing policy $\pi_\theta$ that directly operates on the prompts. Unlike direct score prediction, this approach allows the policy to capture more nuanced information from the prompts and implicitly manage prediction uncertainty. Additionally, optimizing a policy provides the flexibility to dynamically balance exploration and exploitation, allowing the system to discover new, potentially better models for specific queries while continuing to utilize known high-performing models, ultimately improving performance in situations where uncertainty exists.

The routing policy $\pi_\theta$ is formulated as a mapping from a prompt $x \in \mathcal{X}$ to a categorical distribution over the set of $K$ LLMs. A key challenge in learning such a routing policy is dealing with diverse LLM routing candidates. For example, users might specify different sets of models depending on their application needs, or new models could be added after the routing system is deployed. Therefore, the policy must accommodate varied action spaces, where both the number of models and the specific set of available models may change dynamically. To address the varied action space, we condition the routing policy on the set of model identity vectors $\{I_k\}_{k=1}^K$, which represent the capabilities of each model. Conditioning on the action space allows the policy to be aware of all available choices, thereby improving routing decisions even in dynamic environments. Furthermore, we can condition the policy on other available context information, such as the cost $c_k$ and the score prediction $\hat{p}_k = \text{sigmoid}(f(x, I_k))$. These context information can assist the policy to make better-informed routing decisions across different LLM candidates. The policy is defined as

$$\pi_\theta(k' \mid x, \{(I_k, c_k, \hat{p}_k)\}_{k=1}^K) \propto \exp\left(I_{k'}^T h(x, \{(I_k, c_k, \hat{p}_k)\}_{k=1}^K)\right), \tag{5}$$

where $h(\cdot)$ is a neural network that is permutation-invariant with respect to the exchangeable set inputs. The output of $h$ is a vector representation in $\mathbb{R}^d$, and the inner product between the network's output and each model's identity vector determines which model to select according to the policy.

In addition to varied routing candidates, the policy must also incorporate different user preferences. We extend the policy to be conditioned on the preference vector $\boldsymbol{\omega}$, enabling it to tailor decisions based on specific user-defined priorities. Conditioning the routing policy on user preference enhances the policy's ability to generalize to different user scenarios and objectives, allowing for a more personalized and efficient routing mechanism. The routing policy is thus defined as

$$\pi_\theta(k' \mid x, \{(I_k, c_k, \hat{p}_k)\}_{k=1}^K, \boldsymbol{\omega}) \propto \exp\left(I_{k'}^T h(x, \{(I_k, c_k, \hat{p}_k)\}_{k=1}^K, \boldsymbol{\omega})\right). \tag{6}$$

We optimize the policy following standard multi-objective policy gradient algorithms (Xu et al., 2020; Shu et al., 2024), where the gradient for updating the parameters $\theta$ is given by

$$\nabla_\theta [\boldsymbol{\omega}^T \mathbf{J}_{\pi_\theta}] = \mathbb{E}_{x, k'} \left[\boldsymbol{\omega}^T \mathbf{A}(x, k') \nabla_\theta \log \pi_\theta(k' \mid x, \{(I_k, c_k, \hat{p}_k)\}_{k=1}^K, w)\right]. \tag{7}$$

Here, $\mathbf{A}(x, k')$ indicates the advantage function estimated from sampled trajectories. The corresponding value function $\mathbf{V}_{\pi_\theta}(x, \{(I_k, c_k, \hat{p}_k)\}_{k=1}^K)$ outputs a vector of expected returns under the current policy $\pi_\theta$. The parameters of the value function are updated by a squared-error loss $\|\mathbf{V}_{\pi_\theta} - \mathbf{V}_{targ}\|^2$, where $\mathbf{V}_{targ}$ is the target value. Note the value function does not depend on the preference $\boldsymbol{\omega}$, which encourages shared values estimation across different user preferences. The vectorized value function is inspired by the core principles of multi-objective Q-learning algorithms (Yang et al., 2019; Basaklar et al., 2022). This value network and policy gradient extension can be

---

**Algorithm 1** Training Procedure of Preference Conditioned Dynamic Routing

---

**Require:** Model identify vectors $I$, Pretraining steps $T1$, Training steps $T2$
**Require:** Comparison dataset $\mathcal{V}$ for pretraining, Evaluation leaderboard $\mathcal{D}$ for training
**Require:** Evaluation score predictor $f(x, I_k)$, Calibration parameters $\alpha$ and $\beta$
**Require:** Routing policy $\pi_\theta$, Preference range $[\omega_{min}, \omega_{max}]$, RL training procedure $\mathbb{P}$

*## Pretraining Stage*

1: **for** step in $[1, \ldots, T1]$ **do**
2:     sample a batch of pretraining data $(x, (k1, k2), (c1, c2)) \sim \mathcal{V}$
3:     sample a batch of preference $\boldsymbol{\omega} = [1, \omega]$ and $\omega \sim U(\omega_{min}, \omega_{max})$     ▷uniform for cost
4:     compute score predictions $\hat{p}_k = \text{sigmoid}(f(x, I_k))$     ▷auxiliary info to the policy
5:     calibrate the score predictions $\bar{p}_k = \text{sigmoid}(\alpha f(x, I_k) + \beta)$     ▷only used to predict action
6:     normalize scores $\bar{p}_k = \bar{p}_k / \max(\{\bar{p}_k\}_{k \in \{k1, k2\}})$ and costs $\bar{c}_k = c_k / \max(\{c_k\}_{k \in \{k1, k2\}})$
7:     obtain routing action $\hat{a} = \arg\max_{k \in \{k1, k2\}} \boldsymbol{\omega}^T [\bar{p}_k, -\bar{c}_k]$     ▷maximize scalarized reward
8:     pretrain the policy by optimizing $-\log \pi(\hat{a} \mid x, \{(I_k, \bar{c}_k, \hat{p}_k)\}_{k \in \{k1, k2\}}, \boldsymbol{\omega})$
9: **end for**

*## Training Stage*

10: Initialize a replay buffer $\mathbb{B}$     ▷with on-manifold mixup regularization
11: **for** step in $[1, \ldots, T2]$ **do**
12:     sample a batch of training data $(x, \{(M_k, c_k, s_k)\}_{k=1}^K) \sim \mathcal{D}$  ▷$K$ is different across batches
13:     sample a batch of preference $\boldsymbol{\omega} = [1, \omega]$ and $\omega \sim U(\omega_{min}, \omega_{max})$     ▷uniform for cost
14:     normalize scores $\bar{s}_k = s_k / \max(\{s_k\}_{k=1}^K)$ and costs $\bar{c}_k = c_k / \max(\{c_k\}_{k=1}^K)$
15:     compute score predictions $\hat{p}_k = \text{sigmoid}(f(x, I_k))$     ▷auxiliary info to the policy
16:     run the current policy $a \sim \pi(x, \{(I_k, \bar{c}_k, \hat{p}_k)\}_{k=1}^K, \boldsymbol{\omega})$ and obtain reward $[\bar{s}_a, -\bar{c}_a]$
17:     update replay buffer $\mathbb{B} \leftarrow (x, a, \{(M_k, \bar{c}_k, \bar{s}_k)\}_{k=1}^K, \boldsymbol{\omega})$
18:     RL training on data sampled from the replay buffer $\mathbb{P}(\pi_\theta, \mathbb{B})$     ▷with mixup interpolation
19: **end for**

---

seamlessly integrated into most existing policy gradient methods. In our implementation, we adapt Proximal Policy Optimization (PPO) (Schulman et al., 2017), where the clipped surrogate objective is used to update policy parameters. Additionally, Generalized Advantage Estimation (GAE) (Schulman et al., 2015) is employed to compute the advantage function $\mathbf{A}$ and target values $\mathbf{V}_{targ}$. For detailed derivations and implementation specifics, please refer to Appendix A.2.

A key advantages of our preference-conditioned action-space-aware routing policy is its scalability. By utilizing model identity vectors, the policy can seamlessly adapt to diverse routing candidates and incorporate new models without requiring retraining from scratch. As additional LLMs and domains are introduced, the policy efficiently assimilates new information, offering flexibility for future expansion. Additionally, by accounting for user preferences, the policy remains adaptable to varying application requirements and varying computational budgets. However, these benefits are not solely attributed to the policy design. In the following sections, we explore the training methodologies and regularization techniques that ensure the routing policy generalizes effectively across various scenarios.

## 2.5 GENERALIZATION OF THE ROUTING POLICY

In this section, we discuss generalizing the routing policy to handle diverse user preferences, varied routing candidates, and a wide range of prompts, while also addressing the integration of new models into the routing system. Please refer to Algorithm 1 for pseudo code of the training procedure.

**Generalize to Varied Routing Candidates** To ensure the routing policy generalizes to arbitrary sets of LLMs, it is designed to incorporate the action space as an input. To further enhance generalization across different sets of models, we train the policy on diverse trajectories sampled from varied action spaces. In practice, we leverage existing evaluation leaderboards, such as HELM (Liang et al., 2022), where scores are provided for a wide range of models. During training, we randomly select sets of models with differing cost profiles and apply the current policy to route evaluation prompts to the most appropriate LLM. The resulting routing trajectories are stored in a replay buffer, which is used to train the routing policy in an off-policy manner.

**Generalize to Unseen Prompts** A major challenge in designing a routing system is ensuring its generalizability to unseen prompts. While LLM evaluation leaderboards like HELM (Liang et al., 2022) already include diverse prompts, they still fall short of covering all possible user queries encountered in practice. To diversify training data, we propose pretraining the routing policy on pairwise comparison datasets, such as Nectar (Zhu et al., 2023) and Chatbot Arena (Zheng et al., 2023). These datasets feature diverse user queries but lack model-specific evaluation scores $s(x, k)$. Instead, these datasets provide binary winning labels, making them unsuitable for direct policy training. Training the policy to simply predict the winning models may not align with the optimal routing choices, especially when considering cost and user preferences. A straightforward approach might rely on predicted scores $\hat{p}_k = \text{sigmoid}(f(x, I_k))$. However, inaccuracies in these predictions can degrade routing quality. To mitigate this, we propose calibrating the predicted scores using Platt scaling (Platt et al., 1999). Specifically, given the predicted logits $f(x, I_{k1})$ and $f(x, I_{k2})$, we fit a logistic regression model to predict the binary winning label $z \in \{0, 1\}$ with the following probability: $p(z = 1) = \text{sigmoid}(\alpha(f(x, I_{k1}) - f(x, I_{k2})) + \beta)$, where $\alpha$ and $\beta$ are learnable parameters. The calibrated scores are then computed as $\bar{p}_k = \text{sigmoid}(\alpha f(x, I_k) + \beta)$. The routing action is determined by $\hat{a} = \arg\max_{k \in \{k1, k2\}} \boldsymbol{\omega}^T [\bar{p}_k, -c_k]$, and the policy is trained in a supervised manner to predict these actions. By exposing the policy to diverse prompts during pretraining, we encourage it to generalize to unseen queries. An alternative to the supervised pretraining is using calibrated scores within an RL framework, however, preliminary results indicate that RL is more sensitive to prediction errors, therefore we opt for the simpler supervised pretraining.

In addition to large-scale pretraining, research has shown that certain regularization techniques can enhance the generalizability of RL policies (Farebrother et al., 2018; Wang et al., 2020). We adapt the mixup regularization (Zhang, 2017), which has proven effective in RL (Wang et al., 2020), and tailor it to our routing setting. Specifically, we introduce on-manifold mixup, where prompt embeddings are linearly combined with their nearest neighbors from the replay buffer. This neighborhood-based mixup ensures that the interpolated embeddings remain on the manifold of the prompt space. For further details, see Appendix A.3.2.

**Generalize to Diverse User Preferences** To accommodate diverse user preferences, we train the policy using trajectories sampled with varying preferences. The challenge arises from the diversity of routing candidates, as different sets of models exhibit different score and cost scales. As a result, the same preference may lead to different trade-offs across different sets of LLMs. To address this, we normalize the scores and costs so that the highest score and cost in each LLM set equals 1.0. We define the preference vector as $\boldsymbol{\omega} = [1, \omega]$ and sample $\omega$ uniformly from the range $[0, 2]$. In practice, this simple normalization approach has proven effective. We apply the normalization during both the supervised pretraining and RL training stages. Please find further details in Appendix A.3.3.

**Generalize to New Routing Candidates** As LLM development progresses, new models will be added to the routing system. Although the routing policy can handle unseen models, it is crucial to generate a model identity vector for any newly added model, $\tilde{M}$, to capture its unique strengths and weaknesses. This vector can be derived by evaluating the model on a selected set of prompts and optimizing the IRT prediction loss(equation 2) along with the KL loss (equation 4). The optimization process keeps the model's weights fixed while updating only the identity vector, $\tilde{I}$. Given a binarized set of evaluation scores $\tilde{Y}$ on prompts $\tilde{X}$, the model identity vector is computed as:

$$\tilde{I} = \arg\min_I \mathcal{L}_{irt} + \mathcal{L}_{KL} = \arg\min_I \left[ \mathbb{E}_{\tilde{x}} \left[ -\bar{y} \log p - (1 - \bar{y}) \log(1 - p) \right] + D_{\text{KL}} \left( q(\tilde{I}) \| p(\tilde{I}) \right) \right],$$

(8)

where $p = \text{sigmoid}(f(\tilde{e}, \tilde{I}))$ and $\tilde{e}$ is the prompt embedding for prompts $\tilde{x} \in \tilde{X}$. The main challenge is selecting the most informative set of prompts, $\tilde{X}$, to obtain the evaluation scores. Although evaluating the model on all available prompts would be ideal, it is often prohibitively expensive due to high inference costs. To reduce the cost, we propose selecting a subset of prompts through stratified sampling, where the strata are based on the average prediction accuracy of each prompt across existing models. Given a set of prompts $X = \{x_n\}_{n=1}^N$ and evaluation scores $Y_k = \{\bar{y}_{kn}\}_{n=1}^N$ for each available LLM $M_k$, we define the strata as:

$$\psi_n = \mathbb{E}_k \left[ -\bar{y}_{kn} \log p_{kn} - (1 - \bar{y}_{kn}) \log(1 - p_{kn}) \right],$$

(9)

where the expectation is taken over all available LLMs $M_k$. This strata value reflects the difficulty of each evaluation prompt: if most models struggle with a prompt $x_n$, the corresponding strata $\psi_n$ will be high, and if the prompt is easy for most models, the strata value will be lower. Stratified sampling

based on this difficulty ensures that the selected prompts $\tilde{X}$ provide an effective assessment of the new model $\tilde{M}$'s capabilities. While the above method focuses on selecting prompts from LLM evaluation benchmarks, it can also be extended to pairwise comparison datasets. Further details can be found in Appendix A.3.4.

# 3 RELATED WORKS

**LLM Ensemble, Cascade and Routing** As the number of LLMs grows, there is increasing interest in combining them to optimize performance and balance costs. LLM ensemble methods improve response quality by aggregating outputs from multiple LLMs but incur high computational costs since they require running inference on multiple models (Jiang et al., 2023; Wang et al., 2023; Lu et al., 2024). LLM cascading reduces costs by invoking LLMs sequentially, starting with the least expensive model and progressing to more costly ones until a satisfactory response is obtained (Chen et al., 2023; Madaan et al., 2023; Ramírez et al., 2024). While effective in reducing costs, cascading still requires multiple inferences, especially for complex queries, and often depends on an additional model to assess the quality of the responses.

In contrast, LLM routing sends queries directly to the most appropriate model, requiring only a single inference and thus offering a more cost-efficient solution. Typical routing methods rely on performance prediction models to guide the selection of the optimal LLM. These methods either predict downstream evaluation or reward scores for a given query prompt (Shnitzer et al., 2023; Lu et al., 2023; Hari & Thomson, 2023; Šakota et al., 2024), or estimate win rates between pairs of models (Ding et al., 2024; Ong et al., 2024). The chosen LLM is then selected based on predicted performance and any additional constraints, such as cost or latency.

The most relevant work to ours is MetaLLM (Nguyen et al., 2024), which also frames the routing task as a multi-armed bandit problem. However, MetaLLM optimizes a scalarized reward and operates on a fixed set of LLMs, limiting the learned policy to specific user preferences and a predefined set of models. Our approach, by contrast, generalizes to varied user preferences and dynamically adapts to new LLMs added to the system, ensuring broader applicability and greater flexibility.

**Multi-objective Reinforcement Learning** Multi-objective RL seeks to optimize multiple, often conflicting reward signals within a Markov decision process, resulting in a set of Pareto-optimal policies known as the Pareto set rather than a single optimal policy. Traditional algorithms typically aim to approximate this Pareto set by searching for a finite number of policies (Van Moffaert & Nowé, 2014; Parisi et al., 2014; Xu et al., 2020). However, these methods face the curse of dimensionality, where the number of policies needed to accurately approximate the Pareto set grows exponentially with the number of objectives. To address this, recent approaches have proposed using a single deep neural network conditioned on preferences to represent the entire Pareto set (Yang et al., 2019; Abels et al., 2019; Basaklar et al., 2022). Another approach involves using hypernetworks (Chauhan et al., 2023), which map user preferences to the parameters of the policy network (Shu et al., 2024). Our routing policy aligns with the conditional neural network framework, where a single model is conditioned on user preferences to adapt to different user requirements. We further tailor this conditional architecture specifically for routing in LLM systems, allowing for efficient decision-making across a diverse and expanding set of models.

**Generalization in Reinforcement Learning** Generalizing RL policies to new tasks, often referred to as zero-shot RL, is a growing area of research focused on enabling policies to handle unseen tasks without retraining (Korkmaz, 2024). Approaches typically fall into three categories: The first category focuses on maximizing worst-case performance across tasks, often using adversarial training (Moos et al., 2022; Dong et al., 2023). This approach is commonly used when no data is available to identify the current task. The second category aims to compute task representations from data, allowing agents to adapt their policies to the specific task at hand. This approach is commonly employed in multi-task RL and hidden-parameter MDPs (Konidaris & Doshi-Velez, 2014), where task representations are inferred from exploration data within the task environment (Touati & Ollivier, 2021; Agarwal et al., 2021; Benjamins et al., 2022; Ingebrand et al., 2024). The third category leverages in-context learning by feeding data from the current task directly into a pretrained transformer as context (Melo, 2022; Brohan et al., 2022). Although transformers have demonstrated effectiveness, their high memory consumption, training instability, and data inefficiency present challenges to their broader application. Our routing policy falls into the second category, where the task represen-

tation is explicitly provided as a set of LLMs and their associated costs. In a similar vein, Jain et al. (2020) explore RL generalization to new action spaces using a VAE to learn action representations, whereas we capture LLM capabilities via identity vectors.

In addition to task generalization, research has also explored generalizing RL policies to new observation distributions. Techniques include data augmentation (Cobbe et al., 2019; Yarats et al., 2021; Laskin et al., 2020), specialized architectures (Lee et al., 2019b), regularization methods (Farebrother et al., 2018; Wang et al., 2020), invariant representation learning (Tachet et al., 2018; Zhang et al., 2020; Agarwal et al., 2021), and adversarial observation perturbations (Zhang & Guo, 2021; Korkmaz, 2022). Our approach explores a simple regularization technique that encourage smoothness across prompt distributions.

## 4 EXPERIMENTS

In this section, we assess our routing policy on several popular LLM benchmarks, including HELM-Lite (Liang et al., 2022) HELM-MMLU (Liang et al., 2022), HuggingFace OpenLLM Leaderboard (Beeching et al., 2023), HuggingFace OpenLLM Leaderboard v2 (Fourrier et al., 2024), and AlpaceEval 2.0 (Li et al., 2023). We divide the prompts in each leaderboard into training and test splits. The training split is used to train the routing policy, and the test split is reserved to evaluate the routing performance. The routing policy is first pretrained on pairwise comparison datasets, including Chatbot Arena (Zheng et al., 2023), Nectar (Zhu et al., 2023), and a synthetic dataset from RouteLLM (Ong et al., 2024). We train the IRT model on the same pairwise datasets and the training splits of all leaderboards, and we use this IRT model across all evaluations. For cost estimation, we approximate the model invocation costs based on processing and generating 1M tokens each. We evaluate our routing policy across various LLM candidates. Please refer to Appendix B for further details on cost estimation and experimental setup.

Following RouteLLM (Ong et al., 2024), we evaluate our approach in a scenario with two LLM candidates, GPT-4 and Mixtral-8x7B, and compare it to RouteLLM. It is important to note that while RouteLLM is specifically trained for this two-model configuration, our routing model is designed to handle arbitrary sets of LLM candidates. To further test generalization, we evaluate two additional LLM configurations for each dataset. In these multi-LLM settings, RouteLLM is not applicable, as it is restricted to two candidates. For RouteLLM, we adjust the routing preference by specifying different thresholds, whereas our routing models can directly take preferences as inputs. When evaluating scenarios with only two LLM candidates, we also compare against a random baseline, where the model is selected randomly, and the preference is adjusted according to the selection probability. However, in scenarios with more than two LLMs, adjusting the selection probability becomes non-trivial, so we omit the random baseline in these cases. We also compare against a baseline that uses the predicted scores $\hat{p}_k$ to compute utility (denoted as *Predictor*). Another baseline involves training a PPO routing policy using the scalarized reward $r_\omega$, but this requires separate policies for each LLM set and preference. Lastly, we introduce an oracle policy that selects the LLM based on the actual evaluation scores $s(x, k)$.

**Results** Figure 2 shows the routing performance across 5 LLM evaluation leaderboards and various sets of routing candidates. Our *Predictor* baseline already surpasses RouteLLM, demonstrating the strength of our proposed evaluation score prediction model. Notably, our routing policy further exceeds the *Predictor* baseline in scenarios where the predictor may yield inaccurate score predictions. When comparing our routing model to the PPO baseline, we observe that our model achieves comparable or better performance at the same cost across all datasets and LLM configurations, even though PPO is specifically trained for each LLM candidate and preference setting. This highlights the effectiveness of our preference-conditioned, action-space-aware routing policy in generalizing to different LLM candidates and accommodating diverse user preferences. However, a noticeable gap remains between all routing policies and the *Oracle* policy, indicating that there is still room for further improvement.

**Generalize to New Routing Candidates** To simulate the scenario where new models are introduced into the routing system, we select several unseen models from the HuggingFace OpenLLM v2 benchmark. These models are not used for training either the IRT model or the routing policy. For detailed evaluation settings, please refer to Appendix B.8. The identity vectors for these models are obtained by optimize equation 8 over a selected subset of prompts from the OpenLLMv2

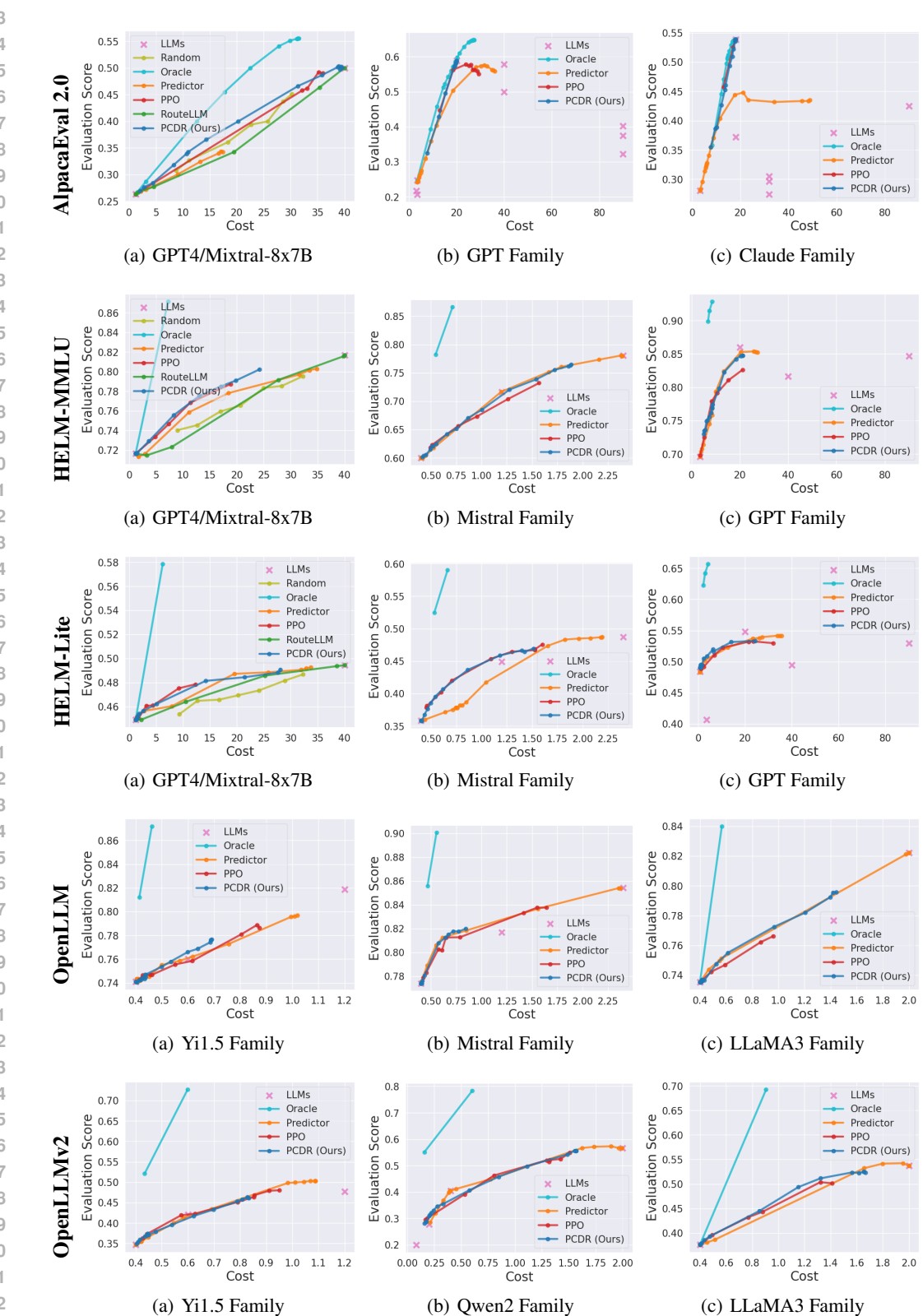

Figure 2: Evaluate the routing performance across 5 datasets and various sets of LLM candidates.

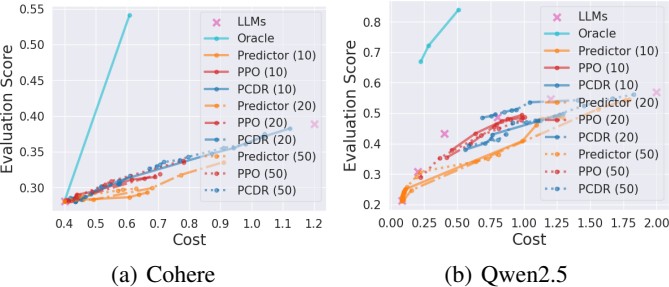 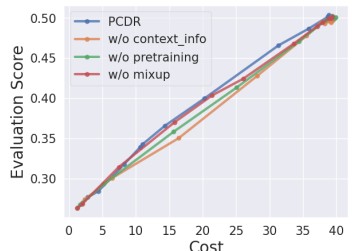

(a) Cohere      (b) Qwen2.5

Figure 3: Evaluate routing performance on two sets of new models. the identity vectors are obtained using 10, 20 or 50 selected prompts, respectively.

Figure 4: Ablation studies on different components of the routing policy.

benchmark. We explore different evaluation budgets, selecting 10, 20 or 50 prompts to obtain the evaluation scores for these newly added models, and then optimize equation 8 with these scores. The *Predictor* baseline utilizes the learned identity vectors to predict the evaluation scores, while the *PPO* baseline trains the routing policy using the same set of selected prompts. For our preference conditioned routing policy, we directly plug the identity vectors into the routing policy trained on OpenLLMv2 (as shown in the last row of Figure 2), without further tuning on these newly added models. Figure 3 presents the evaluation results. Overall, our routing policy outperforms the *Predictor* baseline and performs comparably to the *PPO* policy, despite the latter being specifically trained on the new models. Additionally, our approach proves effective even with a limited number of evaluation scores, enabling efficient onboarding of new routing candidates.

**Ablation Studies** Our routing policy consists of a supervised pretraining stage followed by a RL training stage. During training, we incorporate on-manifold mixup regularization to improve generalization to unseen prompts. Additionally, our policy leverages the predicted scores $\hat{p}_k$ as contextual information. In this section, we perform ablation studies to assess the contributions of these components. Figure 4 presents the results when each component is removed. The results indicate that the context information, pretraining stage, and mixup regularization all contribute to learning a more effective routing policy.

## 5 CONCLUSION

In this work, we introduce a novel routing policy for selecting among LLMs based on user-specific preferences. We frame the problem as a multi-objective optimization task, balancing performance and cost, and proposed a preference-conditioned approach that adapts to individual user preference at inference time. Our method generalized to new and unseen LLMs by leveraging a shared model identity space, enabling seamless integration of new models as they emerge. Through comprehensive experiments on popular evaluation benchmarks, we demonstrate the effectiveness of our routing policy, showing its ability to generalize across a variety of LLMs, prompts and user preferences.

Despite these promising results, there are several areas for future improvement. First, we train the routing policy in an offline setting, where all evaluation scores are pre-computed on a fixed set of prompts. A more dynamic, online setting-where the model adapts based on real-time feedback—would likely improve the robustness and generaliability of the policy. Second, we treat the cost of each model as a constant, but in practice, the cost can vary depending on factors such as input length. Future work could explore adaptive cost modeling that takes into account the specific input characteristics. Additionally, while our focus has been on pure LLM performance, many modern LLMs have the ability to invoke external tools or perform online searches, which could provide richer decision-making capabilities. Incorporating these external functions into the routing policy would be a valuable extension. Finally, although we normalize the scores and costs to align with user preferences, more sophisticated methods for calibrating user preferences could be explored. Setting real-valued preferences may not be intuitive for all users, and designing a more user-friendly interface for preference input, or automating the calibration process, could improve user experience.

In conclusion, our work demonstrates the potential of adaptive, preference-based LLM routing systems in maximizing utility across multiple models. With further advancements, these systems can be made even more robust, efficient, and user-friendly, meeting the evolving demands of LLM usage in real-world applications.

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

# A  METHOD

## A.1  MODEL IDENTITY VECTOR

We learn the model identity vector $I_k$ following a variational variant of the IRT model. Given evaluation scores $Y_k = \{y_{kn}\}_{n=1}^N$ for model $M_k$ on a set of prompts $X = \{x_n\}_{n=1}^N$, we maximize the following variational lower bound of the log-likelihood:

$$
\begin{aligned}
\log p(y_{kn} \mid x_n) &= \log \int p(y_{kn}, I_k \mid x_n) dI_k \\
&\geq \mathbb{E}_{q(I_k)} \left[ \log p(y_{kn} \mid x_n, I_k) \right] - D_{\mathrm{KL}}(q(I_k) \| p(I_k)).
\end{aligned}
\tag{A.1}
$$

Here, the model embedding $I_k$ is treated as a latent variable, with the posterior and prior distributions over $I_k$ denoted by $q(I_k)$ and $p(I_k)$, respectively. In practice, both distributions are modeled as Gaussians, with the posterior $q(I_k) = \mathcal{N}(\mu_k, \Sigma_k)$ and the prior $p(I_k) = \mathcal{N}(\mathbf{0}, \mathbf{I})$. The posterior mean $\mu_k$ and variance $\Sigma_k$ are represented as embedding vectors of dimension $d$, with the variance assumed to be diagonal. The predictive distribution $p(y_{kn} \mid x_n, I_k)$ is implemented as a neural network that concatenates of prompt and model embeddings as input and outputs the score prediction logits. During training, the loss is computed over the entire evaluation benchmarks, involving multiple prompts and models, i.e., $-\mathbb{E}_{x,k} \log p(y_{kn} \mid x_n)$.

### A.1.1  TRAINING WITH REAL-VALUED EVALUATION SCORES

Certain evaluation datasets produce real-valued evaluation scores, such as F1 and RougeL. In order to unify the training procedure, we propose to binarize the real-valued scores. Specifically, given a set of real-valued scores $Y = \{y_n\}_{n=1}^N$, where $y_n \in [0, 1]$, we find an optimal threshold $\eta^*$ so that the average performance across instances are close to the original scores, that is

$$
\eta^* = \arg\min_\eta \left( \frac{1}{N} \sum_{n=1}^N \mathbb{I}(y_n > \eta) - \frac{1}{N} \sum_{n=1}^N y_n \right)^2,
\tag{A.2}
$$

where $\mathbb{I}(y_n > \eta)$ is the indicator function, which equals to 1 only when the condition $y_n > \eta$ is true. Therefore, the binarized evaluation scores are derived as $\bar{Y} = \{\mathbb{I}(y_n > \eta^*)\}_{n=1}^N$.

## A.2  PREFERENCE CONDITIONED ROUTING POLICY

In the main text, we derived the routing policy as

$$
\pi_\theta(k' \mid x, \{(I_k, c_k, \hat{p}_k)\}_{k=1}^K, \boldsymbol{\omega}) \propto I_{k'}^T h(x, \{(I_k, c_k, \hat{p}_k)\}_{k=1}^K, \boldsymbol{\omega}),
$$

where $h(\cdot)$ is a neural network that is permutation invariant to the set $\{(I_k, c_k, \hat{p}_k)\}_{k=1}^K$. We achieve the permutation invariance by using a permutation invariant embeddings of the set, implemented via the SetTransformer architecture(Lee et al., 2019a). The prompt $x$ is encoded using pretrained prompt embeddings. The preference vector $\boldsymbol{\omega}$ is projected through a linear layer for integration into the routing policy. The neural network then concatenates the embeddings and passes them through several linear layers, resulting in a vector representation in $\mathbb{R}^d$. The inner product between $h(\cdot)$ and each model embedding $I_{k'}$ determines which model to select based on the policy. Specifically, the routing probability for selecting model $M_{k'}$ follows the softmax distribution:

$$
\pi_\theta(k' \mid x, \{(I_k, c_k, \hat{p}_k)\}_{k=1}^K, \boldsymbol{\omega}) = \frac{\exp\left(I_{k'}^T h(x, \{(I_k, c_k, \hat{p}_k)\}_{k=1}^K, \boldsymbol{\omega})\right)}{\sum_{k''=1}^K \exp\left(I_{k''}^T h(x, \{(I_k, c_k, \hat{p}_k)\}_{k=1}^K, \boldsymbol{\omega})\right)}.
\tag{A.3}
$$

We train the routing policy following the multi-objective PPO algorithm, where the gradient for updating the policy parameters $\theta$ is given by

$$
\nabla_\theta[\boldsymbol{\omega}^T \mathbf{J}_{\pi_\theta}] = \mathbb{E}_{x,k'} \left[ \boldsymbol{\omega}^T \mathbf{A}(x, k') \nabla_\theta \log \pi_\theta(k' \mid x, \{(I_k, c_k, \hat{p}_k)\}_{k=1}^K, w) \right],
$$

where $\mathbf{A}(x, k')$ indicates the advantage function estimated via GAE (Schulman et al., 2015). The PPO algorithm also requires a value estimation to reduce the gradient variance. Following multi-objective RL literature (Xu et al., 2020; Shu et al., 2024), we define a value network $\mathbf{V}_{\pi_\theta}(x, \{(I_k, c_k, \hat{p}_k)\}_{k=1}^K)$ that outputs a vector of expected returns under the current policy $\pi_\theta$. The value estimation is not conditioned on the preference, therefore, it can be shared across different user preferences. We train the values network by optimizing a MSE loss $\|\mathbf{V}_{\pi_\theta} - \mathbf{V}_{targ}\|^2$, where $\mathbf{V}_{targ}$ indicates the target values estimated via GAE.

### A.3 GENERALIZATION OF THE ROUTING POLICY

In this section, we discuss the training procedure of the dynamic routing policy, which is designed to enhance the generalizability of the policy to various scenarios.

#### A.3.1 SUPERVISED PRETRAINING

The supervised pretraining stage leverages diverse prompts from pairwise comparison datasets to enhance generalization to unseen prompts. Given a pairwise comparison dataset $\mathcal{V}$, where each example consists of a prompt $x_n$, a pair of models $M_{k1}$ and $M_{k2}$, and a winning label $z_n \in \{0, 1\}$, we first train a logistic regression model to calibrate the predicted evaluation scores, $\hat{p} = \text{sigmoid}(f(x, I_k))$, using the winning label $z_n$. Specifically, the logical regression model predicts the wining probability as $p(z_n = 1) = \text{sigmoid}(\alpha(f(x_n, I_{k1}) - f(x_n, I_{k2})) + \beta)$, where $\alpha$ and $\beta$ are learnable parameters. After training, the calibrated evaluation scores are given by $\bar{p} = \text{sigmoid}(\alpha f(x, I_k) + \beta)$. The calibration follows the well-known Platt scaling (Platt et al., 1999) algorithm, which refines the evaluation scores using human-labeled winning labels to produce more accurate predictions.

With the calibrated evaluation scores $\bar{p}$ on a prompt $x$ and a user preference vector $\boldsymbol{\omega}$, the routing action is determined by $\hat{a} = \arg\max_{k \in \{k1, k2\}} \boldsymbol{\omega}^T[\bar{p}_k, -c_k]$. We then pretrain the routing policy in a supervised manner using the following negative log-likelihood loss:

$$\mathcal{L}_{pretrain} = -\log \pi(\hat{a} \mid x, \{(I_k, c_k, \hat{p}_k)\}_{k \in \{k1, k2\}}, \boldsymbol{\omega}). \tag{A.4}$$

It is important to note that the policy utilizes the original predicted scores $\hat{p}$ as input, rather than the calibrated scores, to maintain consistency with the subsequent RL training stage.

#### A.3.2 ON-MANIFOLD MIXUP REGULARIZATION

The mixup regularization technique was initially introduced for supervised learning tasks (Zhang, 2017), where new input-output pairs are generated by taking convex combinations of pairs of training samples. Wang et al. (2020) extended this approach to RL, where observations and their associated supervision signals from two transitions are combined convexly. In our case, the observation corresponds to the prompt embeddings. However, naively combining two prompt embeddings may produce vectors that lie outside the prompt manifold. To address this, we use the nearest neighbor from the replay buffer for each prompt $x$. Given the embedding $e$ for prompt $x$ and the embedding $e_n$ for its nearest neighbor, the interpolated prompt embedding is obtained as:

$$\hat{e} = \lambda e + (1 - \lambda)e_n, \tag{A.5}$$

where $\lambda \sim \text{Beta}(\xi, \xi)$, and $\xi$ is a hyperparameter, set to 0.2 as recommended in the original mixup paper. To train the routing policy on the interpolated prompt embeddings using PPO, we similarly interpolate the associated supervision signals:

$$\hat{\pi}_{old} = \lambda \pi_{old} + (1 - \lambda)\pi_{old}^{(n)}$$
$$\hat{\mathbf{A}} = \lambda \mathbf{A} + (1 - \lambda)\mathbf{A}_n \tag{A.6}$$
$$\hat{\mathbf{V}}_{targ} = \lambda \mathbf{V}_{targ} + (1 - \lambda)\mathbf{V}_{targ}^{(n)}$$

The interpolated routing action $\hat{a}$ is chosen as $a$ if $\lambda > 0.5$, otherwise $a_n$. Similarly, routing-relevant parameters, including $I_k$ and $\boldsymbol{\omega}$ are chosen based on $\lambda$ as well.

#### A.3.3 REWARD NORMALIZATION

Our routing policy is designed to generalize across different sets of LLM candidates. However, the varying score and cost scales across these sets can pose challenges. For instance, routing decisions involving proprietary API models often involve higher costs compared to open-source models, where the cost is significantly lower. These discrepancies in scale can complicate the training of the routing policy, as the preference vector must be adjusted to suit each scenario. Moreover, the same preference vector might favor higher costs for one set of models while preferring lower costs for another, introducing inconsistency and instability during training. To address this, we propose normalizing both the scores and costs across all LLM sets. Given a set of LLMs $\{M_k\}_{k=1}^K$ with scores $\{s_k\}_{k=1}^K$ and costs $\{c_k\}_{k=1}^K$, we normalize the scores and costs by

$$\bar{s}_k = s_k / \max(\{s_k\}_{k=1}^K), \quad \bar{c}_k = c_k / \max(\{c_k\}_{k=1}^K). \tag{A.7}$$

This normalization ensures that both scores and costs are scaled such that their maximum value is 1.0. By standardizing the range of values, the policy can learn a consistent mapping from user preferences to routing decisions across various LLM sets. This approach prevents the policy from disproportionately favoring either high-cost or low-cost models based purely on their relative scales, promoting more balanced decisions that accurately reflect trade-offs between performance and cost.

In theory, the preference vector $\boldsymbol{\omega}$ can take any value in the range of $[0, \infty)$. However, for simplicity, we define it as $\boldsymbol{\omega} = [1, \omega]$, fixing the preference weight for scores at 1 and only varying the weight for cost. When $\omega = 0$, the model selection prioritizes high scores regardless of cost, while $\omega = \infty$ indicates a preference for the lowest-cost model. In practice, we found that sampling $\omega$ from the range $[0, 2]$ effectively captures the Pareto front.

### A.3.4 STRATIFIED SAMPLING

Generalizing the routing policy to a new model $\tilde{M}$ requires to obtain its identity vector $\tilde{I}$, which captures the model's unique strengths and weaknesses. However, evaluating the model on all available prompts is often prohibitively expensive, especially when new models are frequently introduced. In order to reduce the evaluation cost, we propose selecting a subset of informative prompts that effectively assess the model's capabilities. Specifically, given a set of prompts $X = \{x_n\}_{n=1}^N$ and the binarized evaluation scores $Y_k = \{\bar{y}_{kn}\}_{n=1}^N$ for each available LLM $M_k$, we assess the difficulty of each prompt based on the average prediction accuracy across all models $M_k$, i.e.,

$$\psi_n = \mathbb{E}_k \left[ -\bar{y}_{kn} \log p_{kn} - (1 - \bar{y}_{kn}) \log(1 - p_{kn}) \right].$$

We then apply stratified sampling using the difficulty $\psi_n$ as the strata. The stratified sampling ensures the selected prompts covers a range of difficulties, from easy to hard, providing a more balanced and informative assessment of the model's strengths and weaknesses. Once the subsets $\tilde{X}$ is selected, the model identity vector is computed as:

$$\tilde{I} = \arg\min_I \mathcal{L}_{irt} + \mathcal{L}_{KL} = \arg\min_I \left[ \mathbb{E}_{\tilde{x}} \left[ -\bar{y} \log p - (1 - \bar{y}) \log(1 - p) \right] + D_{\mathrm{KL}} \left( q(\tilde{I}) \| p(\tilde{I}) \right) \right],$$

where $p = \mathrm{sigmoid}(f(\tilde{e}, \tilde{I}))$, and $\tilde{e}$ is the prompt embedding for prompts $\tilde{x} \in \tilde{X}$.

The stratified sampling approach described above can also be extended to sample prompts from pairwise comparison datasets. Given a pairwise comparison dataset $\mathcal{V}$, where each example consists of a prompt $x_n$, a pair of models $M_{k1}$ and $M_{k2}$, and a winning label $z_n \in \{0, 1\}$. We first assess each model's capability using Elo score (Elo, 1967). The Elo scores are then used as strata to sample a set of models as the comparison baselines. For each baseline $M_k$, we uniformly select a set of prompts $X_k$ on which to run inference with the new model $\tilde{M}$ and compare its performance to the baseline $M_k$. After obtaining the baseline models and pairwise comparison labels, the model identity vector is computed as:

$$\tilde{I} = \arg\min_I \mathcal{L}_{pair} + \mathcal{L}_{KL}. \tag{A.8}$$

In our experiments, we opted to sample prompts from existing evaluation benchmarks for simplicity. We leave the exploration of sampling from pairwise comparison datasets as future work.

## B EXPERIMENT

### B.1 MODEL COST

In Table B.1, we list the costs for each model. For proprietary APIs, the costs are based on their official API pricing, while for open-source models, we reference pricing from TogetherAI[1]. All costs are normalized by estimating the expense of processing 1 million input tokens and generating 1 million output tokens.

### B.2 DATASET STATISTICS

Our framework consists of three training stages: First, we train the IRT model to obtain model identity vectors $I$ and the evaluation score prediction model $f$. Second, we perform supervised

---

[1]https://www.together.ai/pricing

Table B.1: The estimated cost of invoking the models for processing 1M input tokens and generating 1M output tokens.

| Model | Cost ($) |
|---|---|
| gpt-3.5-turbo-0125 | 2 |
| gpt-3.5-turbo-0301 | 3.5 |
| gpt-3.5-turbo-0613 | 3.5 |
| gpt-3.5-turbo-1106 | 3 |
| gpt-4-0125-preview | 40 |
| gpt-4o-2024-05-13 | 20 |
| gpt-4o-mini-2024-07-18 | 0.75 |
| gpt-4 | 90 |
| gpt-4-1106-preview | 40 |
| gpt-4-turbo-2024-04-09 | 40 |
| gpt-4-turbo | 40 |
| claude-3-opus | 90 |
| claude-3.5-sonnet | 18 |
| claude-3-sonnet | 18 |
| claude-3-haiku | 1.5 |
| claude-2.1 | 32 |
| claude-2 | 32 |
| claude-instant | 3.2 |
| claude-1 | 32 |
| gemini-pro-1.5 | 14 |
| gemini-flash-1.5 | 0.375 |
| llama3.1-405b | 9 |
| llama3.1-70b | 1.584 |
| llama3.1-8b | 0.324 |
| llama3-70b | 1.584 |
| llama3-8b | 0.324 |
| mistral-large | 12 |
| mistral-medium | 10.8 |
| mistral-small | 8 |
| mixtral-8x22b | 2.16 |
| mixtral-8x7b | 1.08 |
| mixtral-7b | 0.36 |
| command-r-plus | 18 |
| command-r | 2 |
| command | 3 |
| command-light | 0.9 |
| qwen-1.5-110b | 3.24 |
| qwen-1.5-72b | 1.62 |
| yi-large | 6 |

Table B.2: Two types of datasets used in the training process.

| Category | Dataset | # Prompts | # Models |
|---|---|---|---|
| Pairwise Model Comparison | berkeley-nest/Nectar[2] | 182954 | 39 |
| | lmsys/lmsys-arena-human-preference-55k[3] | 39716 | 64 |
| | lmsys/chatbot_arena_conversations[4] | 18320 | 20 |
| | lmsys/mt_bench_human_judgments[5] | 894 | 6 |
| | routellm/gpt4_judge_battles[6] | 84864 | 2 |
| Single Model Evaluation | AlpacaEval 2.0[7] | 805 | 61 |
| | HELM-Lite[8] | 13021 | 61 |
| | HELM-MMLU[9] | 14042 | 45 |
| | OpenLLM Leaderboard[10] | 14617 | 41 |
| | OpenLLM Leaderboard v2[11] | 21606 | 39 |

pretraining of the routing policy on diverse prompts. Third, we train the routing policy using a reinforcement learning procedure. Below, we summarize the datasets used in each training stage.

The datasets used in this work fall into two categories: First, pairwise comparison datasets, where annotations indicate which of two models provides a higher-quality response. Second, LLM evaluation datasets, which provide evaluation scores for various models on a set of prompts. Table B.2 summarizes the statistics of these two types of datasets. We apply basic preprocessing, such as removing multi-turn prompts and excluding ties from pairwise comparisons. For LLM evaluation benchmarks, we select a subset of popular LLMs. Please see Table B.3 for the full list of LLMs involved in this work.

The IRT model is trained using the pairwise comparison datasets and the training splits of the evaluation datasets. The pretraining stage also uses these pairwise comparison datasets. For the policy training stage, the routing policy is trained separately on each LLM evaluation dataset. We do not train the policy across different evaluation benchmarks, as they employ different scoring mechanisms, leading to variations in score scales.

## B.3 TRAINING THE IRT MODEL

The IRT model for evaluation outcome prediction consist of four component: the prompt representation $e$, the model identity vector $I$, the evaluation score predictor $f(e, I)$, and the pairwise winner predictor $g(e, I)$. The prompt representation $e$ is obtained using a pretrained text embedding model, meaning it contains no learnable parameters. The model identity vector is initialized as random embeddings for each model listed in Table B.3, with the embedding dimension set to 128. The two neural networks, $f$ and $g$, share a common backbone, differing only in their final linear layer. This shared architecture encourages the model identity vector to capture both types of evaluation outcomes, enabling more accurate representation of each model's strengths and weaknesses.

The IRT model is trained using both the pairwise comparison datasets and the training splits of the evaluation datasets, with a combined loss function, $\mathcal{L}_{irt} + \mathcal{L}_{pair}$. The model is trained for 10 epochs with a batch size of 256. We use the Adam optimizer with a learning rate of 0.001. The learning rate is decayed by 0.95 after each epoch. We did not conduct extensive hyperparameter tuning, and

---

[2] https://huggingface.co/datasets/berkeley-nest/Nectar

[3] https://huggingface.co/datasets/lmsys/lmsys-arena-human-preference-55k

[4] https://huggingface.co/datasets/lmsys/chatbot_arena_conversations

[5] https://huggingface.co/datasets/lmsys/mt_bench_human_judgments

[6] https://huggingface.co/datasets/routellm/gpt4_judge_battles

[7] https://tatsu-lab.github.io/alpaca_eval/

[8] https://crfm.stanford.edu/helm/lite/latest/

[9] https://crfm.stanford.edu/helm/mmlu/latest/

[10] https://huggingface.co/spaces/open-llm-leaderboard-old/open_llm_leaderboard

[11] https://huggingface.co/spaces/open-llm-leaderboard/open_llm_leaderboard

Table B.3: The models used in this work for training and evaluating the routing policy.

| | | | |
|---|---|---|---|
| ai21_j2-grande | ai21_j2-jumbo | ai21_jamba-instruct | alpaca-7b |
| alpaca-13b | chatglm-6b | chatglm2-6b | chatglm3-6b |
| claude-1 | claude-2.0 | claude-2.1 | claude-instant-1 |
| claude-instant-1.2 | claude-3-5-sonnet-20240620 | claude-3-opus-20240229 | claude-3-sonnet-20240229 |
| claude-3-haiku-20240307 | cohere_command-r | cohere_command | cohere_command-r-plus |
| cohere_command-light | cohere_command-xlarge | codellama-7b-instruct | codellama-13b-instruct |
| codellama-34b-instruct | codellama-70b-instruct | deepseek-llm-67b-chat | dolly-v2-12b |
| dolphin-2.2.1-mistral-7b | dialogpt-large | falcon-180b-chat | falcon-40b-instruct |
| falcon-7b-instruct | fastchat-t5-3b | flat-t5-small | gemini-1.0-pro |
| gemini-1.5-pro | gemini-1.5-flash | gemini-pro-dev-api | gemma-2-9b-it |
| gemma-2-27b-it | gemma-2b-it | gemma-7b-it | recurrentgemma-2b-it |
| recurrentgemma-9b-it | google-text-unicorn | google-text-bison | gpt2 |
| gpt2-large | gpt2-medium | gpt2-xl | gpt-3.5-turbo-0125 |
| gpt-3.5-turbo-0314 | gpt-3.5-turbo-0613 | gpt-3.5-turbo-1106 | gpt-4-0125-preview |
| gpt-4 | gpt-4-0314 | gpt-4-0613 | gpt-4-1106-preview |
| gpt-4-turbo-2024-04-09 | gpt-4o-2024-05-13 | gpt-4o-mini-2024-07-18 | gpt4all-13b-snoozy |
| guanaco-13b | guanaco-33b | guanaco-65b | guanaco-7b |
| koala-13b | llama-13b | llama-65b | llama-2-13b-chat |
| llama-2-70b-chat | llama-2-7b-chat | llama2-70b-steerlm-chat | llama-3-70b-instruct |
| llama-3-8b-instruct | llama-3.1-405b-instruct-turbo | llama-3.1-70b-instruct-turbo | llama-3.1-8b-instruct-turbo |
| luminous-base | luminous-supreme | luminous-extended | mamba-gpt-7b-v2 |
| metamath-13b | metamath-70b | mistral-7b-instruct-v0.1 | mistral-7b-instruct-v0.2 |
| mistral-7b-instruct-v0.3 | mistral-large | mistral-medium | mistral-small |
| mixtral-8x7b-instruct-v0.1 | mixtral-8x22b-instruct-v0.1 | mpt-30b-chat | mpt-7b-chat |
| nous-hermes-2-mixtral-8x7b-dpo | oasst-pythia-12b | opt-1.3b | opt-2.7b |
| opt-350m | opt-6.7b | opt-iml-max-1.3b | opt-iml-max-30b |
| openchat-3.5 | openchat-3.5-0106 | openhermes-2.5-mistral-7b | phi-2 |
| phi-2-dpo | phi-2-sft | phi-3-medium | phi-3-small |
| phi-3-mini | palm-2 | pythia-12b | pplx-70b-online |
| pplx-7b-online | palmyra-x-v3 | palmyra-x-v2 | qwen-14b-chat |
| qwen1.5-0.5b-chat | qwen1.5-1.8b-chat | qwen1.5-4b-chat | qwen1.5-72b-chat |
| qwen1.5-14b-chat | qwen1.5-32b-chat | qwen1.5-7b-chat | qwen1.5-110b-chat |
| qwen1.5-moe-a2.7b-chat | qwen2-0.5b-instruct | qwen2-1.5b-instruct | qwen2-7b-instruct |
| qwen2-72b-instruct | rwkv-4-raven-1b5 | rwkv-4-raven-3b | rwkv-4-raven-7b |
| rwkv-4-raven-14b | solar-10.7b-instruct-v1.0 | stablelm-tuned-alpha-7b | starling-lm-7b-alpha |
| stripedhyena-nous-7b | text_davinci_001 | text_davinci_002 | text_davinci_003 |
| tulu-2-dpo-7b | tulu-2-dpo-13b | tulu-2-dpo-70b | ultralm-13b |
| ultralm-65b | vicuna-13b | vicuna-33b | vicuna-7b |
| wizardlm-7b | wizardlm-13b | wizardlm-70b | yi-6b-chat |
| yi-34b-chat | yi-large | yi1.5-6b-chat | yi1.5-9b-chat |
| yi1.5-34b-chat | zephyr-7b-alpha | zephyr-7b-beta | |

no signs of overfitting were observed during preliminary experiments. Additionally, training beyond 10 epochs did not lead to further improvements in validation performance and downstream routing performance.

### B.4 SUPERVISED PRETRAINING OF THE ROUTING POLICY

The supervised pretraining stage for the routing policy optimizes the following negative log-likelihood $-\log \pi(\hat{a} \mid x, \{(I_k, \bar{c}_k, \hat{p}_k)\}_{k \in \{k1, k2\}}, \boldsymbol{\omega})$ using the pairwise comparison dataset $\mathcal{V}$. Given two models $M_{k1}$ and $M_{k2}$ compared on the prompt $x$, we first sample a preference vector $\boldsymbol{\omega} = [1, \omega]$, where $\omega$ is uniformly sampled from the predefined distribution $U(\omega_{min}, \omega_{max})$. The routing decision is then estimated as $\hat{a} = \arg\max_{k \in \{k1, k2\}} \boldsymbol{\omega}^T[\bar{p}_k, -\bar{c}_k]$, where $\bar{c}_k$ represents the normalized cost $\bar{c}_k = c_k / \max(c_{k1}, c_{k2})$, and $\bar{p}_k$ represent the calibrated evaluation score $\bar{p}_k = \text{sigmoid}(\alpha f(x, I_k) + \beta)$. The evaluation scores are further normalized by dividing by the maximum calibrated scores, ensuring consistency with the scale used in the RL training stage. Note that these calibrated and normalized scores are used only for routing action estimation during pretraining, the policy takes in the original score prediction $\hat{p}_k = \text{sigmoid}(f(x, I_k))$ as auxiliary inputs, since the normalized scores are not available during the test phase.

The pretraining stage runs for 500 steps with a batch size of 1024, using the Adam optimizer with a learning rate of 0.001. The calibration parameters, $\alpha$ and $\beta$, are learned by fitting a logistic regression model. Again, we did not conduct extensive hyperparameter tuning, further tuning may improve the performance.

### B.5 RL TRAINING OF THE ROUTING POLICY

The RL training stage follows a modified PPO procedure tailored for the multi-objective optimization task. Specifically, from the evaluation leaderboard $\mathcal{D}$, we sample $K$ models, $\{M_k\}_{k=1}^{K}$, as routing candidates. The costs $c_k$ of these models are normalized by $\bar{c}_k = c_k / \max(\{c_k\}_{k=1}^{K})$, and the their evaluation scores $s_k$ are normalized by $\bar{s}_k = s_k / \max(\{s_k\}_{k=1}^{K})$. The user preference $\boldsymbol{\omega} = [1, \omega]$ and $\omega$ is sampled from the distribution $U(\omega_{min}, \omega_{max})$. The training process starts with generating the trajectories following the current policy $a \sim \pi(x, \{(I_k, \bar{c}_k, \hat{p}_k)\}_{k=1}^{K}, \boldsymbol{\omega})$. The multi-objective reward for action $a$ is represented as a vector $[\bar{s}_a, -\bar{c}_a]$. We update the replay buffer with these sampled trajectories and use samples from the buffer to train the policy. For mixup regularization, we identify the nearest neighbor for each sampled prompt and perform a weighted linear combination of the prompt embedding and its neighbors, where the weights are drawn from $\text{Beta}(0.2, 0.2)$. Both the reward and the advantage are linearly combined using the same weights.

At each training step, we sample 256 new prompts, along with their routing candidates and preference vectors, to obtain the routing trajectories and update the replay buffer. The training stage runs for 500 steps with a batch size of 256, using Adam optimizer with learning rate of 0.001.

### B.6 EVALUATION SETUP

We evaluate the routing performance on 5 LLM evaluation benchmarks and various sets of routing candidates. Table B.4 presents the detailed evaluation settings.

### B.7 BASELINES

In this section, we describe the implementation details of the baseline methods.

#### B.7.1 ROUTELLM

RouteLLM (Ong et al., 2024) develops a model that predicts the winning label between a pair of LLMs and selects the model based on a threshold applied to the predicted probability. To account for varying user preferences, we evaluate RouteLLM using a range of different thresholds.

Table B.4: Evaluation settings.

| Benchmark | Setting | Models |
|---|---|---|
| AlpacaEval 2.0 | GPT4/Mixtral-8x7B | gpt4_1106_preview
Mixtral-8x7B-Instruct-v0.1 |
| | GPT Family | gpt-3.5-turbo-0301
gpt-3.5-turbo-0613
gpt-3.5-turbo-1106
gpt-4-0125-preview
gpt-4o-2024-05-13
gpt4
gpt4_0314
gpt4_0613
gpt4_1106_preview |
| | Claude Family | claude
claude-2
claude-2.1
claude-3-5-sonnet-20240620
claude-3-opus-20240229
claude-3-sonnet-20240229
claude-instant-1.2 |
| HELM-MMLU | GPT4/Mixtral-8x7B | gpt4_1106_preview
mixtral-8x7b-32kseqlen |
| | Mistral Family | mistral-7b-instruct-v0.3
mixtral-8x22b
mixtral-8x7b-32kseqlen |
| | GPT Family | gpt-3.5-turbo-0613
gpt-4-0613
gpt-4-1106-preview
gpt-4o-2024-05-13 |
| HELM-Lite | GPT4/Mixtral-8x7B | gpt4_1106_preview
mixtral-8x7b-32kseqlen |
| | Mistral Family | mistral-7b-instruct-v0.3
mixtral-8x7b-32kseqlen
mixtral-8x22b |
| | GPT Family | gpt-4o-2024-05-13
gpt-4o-mini-2024-07-18
gpt-3.5-turbo-0613
gpt-4-0613
gpt-4-1106-preview |
| OpenLLM | Yi1.5 Family | Yi-1.5-34B-Chat
Yi-1.5-6B-Chat
Yi-1.5-9B-Chat |
| | Mistral Family | Mistral-7B-Instruct-v0.2
Mixtral-8x22B-Instruct-v0.1
Mixtral-8x7B-Instruct-v0.1 |
| | LLaMA3 Family | Llama-3-70B-Instruct
Llama-3-8B-Instruct |
| OpenLLMv2 | Yi1.5 Family | Yi-1.5-34B-Chat
Yi-1.5-6B-Chat
Yi-1.5-9B-Chat |
| | Qwen2 Family | Qwen2-0.5B-Instruct
Qwen2-1.5B-Instruct
Qwen2-72B-Instruct
Qwen2-7B-Instruct |
| | LLaMA3 Family | Llama-3-70B-Instruct
Llama-3-8B-Instruct |

Table B.5: Evaluation setting fro new routing candidates.

| Benchmark | Setting | Models |
|-----------|---------|--------|
| OpenLLMv2 | Cohere | aya-23-35B |
|           |        | aya-23-8B |
|           | Qwen2.5 | Qwen2.5-0.5B-Instruct |
|           |        | Qwen2.5-1.5B-Instruct |
|           |        | Qwen2.5-7B-Instruct |
|           |        | Qwen2.5-14B-Instruct |
|           |        | Qwen2.5-32B-Instruct |
|           |        | Qwen2.5-72B-Instruct |

### B.7.2 PREDICTOR

The predicted evaluation scores $\hat{p}_k = \text{sigmoid}(f(x, I_k))$ can be used directly to compute the scalarized reward for an LLM $M_k$ as $r_{\boldsymbol{\omega}}(x, k) = \boldsymbol{\omega}^T[\hat{p}_k, -c_k]$. The routing decision is then made by selecting $\hat{a} = \arg\max_k r_{\boldsymbol{\omega}}(x, k)$.

### B.7.3 RANDOM

The random routing policy selects models based on predefined probabilities for each model. Different user preferences are reflected by adjusting these probabilities. However, when there are more than two LLM candidates, specifying the probabilities becomes non-trivial, so we omit the random baseline in these scenarios.

### B.7.4 ORACLE

The oracle routing policy selects the model based on the actual evaluation scores, making the routing decision as $\hat{a} = \arg\max_k r_{\boldsymbol{\omega}}(x, k) = \arg\max_k \boldsymbol{\omega}^T[s(x, k), -c_k]$, where $s(x, k)$ represents the true performance score for model $M_k$ on prompt $x$.

### B.7.5 PPO

For each LLM candidate set and each user preference, we train a separate PPO routing policy to maximize the scalarized reward $r_{\boldsymbol{\omega}}(x, k) = \boldsymbol{\omega}^T[s(x, k), -c_k]$.

### B.8 GENERALIZE TO NEW ROUTING CANDIDATES

To simulate the scenario where new models are introduced into the routing system, we select several unseen models from the HuggingFace OpenLLM v2 benchmark. These models are not used for training either the IRT model or the routing policy. Table B.5 shows the detailed evaluation settings.

