# OpenReview forum: "LLM Bandit: Cost-Efficient LLM Generation via Preference-Conditioned Dynamic Routing"
_ICLR.cc/2025/Conference — Submitted to ICLR 2025_

### Official Review · Reviewer_Zngd · 2024-10-19

**Soundness:** 3
**Presentation:** 1
**Contribution:** 2
**Rating:** 3
**Confidence:** 3

**Summary:**

This paper proposes a method to choose a suitable large language model given a prompt. The proposed method frames the problem as a bandit and then trains a representation for prompts and models. Then a utility function is trained to estimate the reward by calling a specific LLM on a specific prompt. Based on the score and users' preference a LLM is selected. Experiments show that the proposed method is effective and efficient.

**Strengths:**

The strengthens of this paper are listed as follows

1. The studied problem is crucial. Framing the model selection task as a bandit is also interesting

2. The experiment, which includes a wide range of models and datasets, is comprehensive

3. The proposed algorithm is simple yet effective as shown by the experiment

**Weaknesses:**

I have several concerns regarding the clarity, structure, and conceptual alignment of the paper, which are outlined as follows.

1. Clarity of Writing in Section 2: Section 2, particularly Section 2.3, was challenging to follow. There are several points where the explanation is incomplete or ambiguous:

- In Equation (2), it is unclear which parameter is being trained—whether it is $I_k$ or $f$.

- The distribution $p$ and $q$ in equation (4) is not defined, and it is unclear why $I_k$ should be sampled from some dstribution

- In Equation (5), the definition of $I_k^T$ is missing.

Furthermore, Algorithm 1 describes the method in two stages: pretraining and training. However, these terms are not clearly defined in the main text, leaving their motivation unclear. It would be helpful if the author elaborated on the specifics of these stages and how they relate to the overall algorithm.

2. Bandit Problem Formulation: The choice to frame the problem as a bandit problem seems misaligned with the algorithm itself. When I encounter the term "bandit," I expect the use of specific online bandit algorithms or techniques. However, Algorithm 1 appears more akin to a supervised-learning approach, as it seems to use prompt-reward pairs to train the IRT model ($f$) and the model identity vector ($I_k$). So what is the motivation for framing this task as a bandit problem? It would be helpful if the author could more explicitly justify the bandit analogy.

3. Contribution elaboration: Some of the claimed contributions are not well-articulated or explained in the paper. For instance:

- In the second contribution, the author says they proposed a preference-conditioned mechanism. However there is no clear definition or explanation of what preference-conditioned means in the context of the paper.

- Similarly, the term "action-space aware" in the third contribution is introduced but not sufficiently explained. This lack of clarity hinders understanding and prevents a full appreciation of the contribution. A more thorough explanation of these terms and their relevance to the method would be beneficial.

4. In Contribution 4, the author claims that the method generalizes across prompts by "leveraging a wide range of data." However, generalization typically refers to a model's ability to perform well on unseen data—not merely data it has been trained on. Therefore the usage of "generalization" here might not be proper.

**Questions:**

See weakness above

---

> ### Author Response · Authors · 2024-11-16
>
> Thank you for your detailed review and for raising several important points. We appreciate your feedback, as it will help us improve the presentation and ensure a clear understanding of our contributions. Please allow us to clarify the sections you highlighted:
>
> > Clarity of Writing in Section 2:
>
> * In equation 2, the model identity vector $I_k$ and the the neural network $f$ are jointly trained.
> * In equation 4, the distributions $q(I_k)$ and $p(I_k)$ represent the variational posterior and prior distributions, respectively, over the model identity vectors. This is similar to a typical VAE formulation, and here we parameterize $q(I_k)$ and $p(I_k)$ as Gaussian distributions. The KL divergence between posterior ad prior distribution acts as a regularization term on the learned representations so that it can generalize to unseen models.
> * In equation 5, $I_{k^{\prime}}^T$ is the transpose of the model identity vector $I_{k^{\prime}}$. The policy $\pi(k^{\prime} \mid \cdot)$ outputs the probability of selecting model $M_{k^{\prime}}$.
>
>
>
>  > Furthermore, Algorithm 1 describes the method in two stages: pretraining and training. However, these terms are not clearly defined in the main text, leaving their motivation unclear. It would be helpful if the author elaborated on the specifics of these stages and how they relate to the overall algorithm.
>
>
> The pretraining stage is meant to pretrain the routing policy on the large collection of prompts using existing pairwise comparison datasets. Please see details in line 270-288. The training stage is RL training for the routing policy following equation 7.
>
>
>
> > Bandit Problem Formulation: The choice to frame the problem as a bandit problem seems misaligned with the algorithm itself. When I encounter the term "bandit," I expect the use of specific online bandit algorithms or techniques. However, Algorithm 1 appears more akin to a supervised-learning approach, as it seems to use prompt-reward pairs to train the IRT model (f) and the model identity vector (I_k). So what is the motivation for framing this task as a bandit problem? It would be helpful if the author could more explicitly justify the bandit analogy.
>
>
>
> First, let us clarify that Algorithm 1 is for training the routing policy $\pi$, not for training the model identity vectors $I_k$ and IRT model $f$. Second, the motivation for framing the routing task as a multi-armed bandit is that the routing policy will choose one LLM to send the query. Third, the pretraining stage in Algorithm 1 is supervised training, but the training stage follow the RL optimization procedure.
>
>
>
> > In the second contribution, the author says they proposed a preference-conditioned mechanism. However there is no clear definition or explanation of what preference-conditioned means in the context of the paper.
>
>
> The preference conditioned mechanism is explained in line199-205. The motivation to condition the routing policy on user preference is to allow users to specify preference at test time.
>
>
> > Similarly, the term "action-space aware" in the third contribution is introduced but not sufficiently explained. This lack of clarity hinders understanding and prevents a full appreciation of the contribution. A more thorough explanation of these terms and their relevance to the method would be beneficial.
>
>
> The explanation for “action-space aware'” is around line182-198. Since user may specify an arbitrary set of LLMs to be selected at test time, the routing policy needs to adapt to arbitrary action spaces. We accomplish this by condition the routing policy on the set of model identity vectors, thus making the policy aware of the available actions.
>
>
> > In Contribution 4, the author claims that the method generalizes across prompts by "leveraging a wide range of data." However, generalization typically refers to a model's ability to perform well on unseen data—not merely data it has been trained on. Therefore the usage of "generalization" here might not be proper.
>
>
> Thanks for pointing it out, we will rephrase it to make it clear. What we really mean is training on a wide range of data helps the routing policy gain generalizability to unseen prompts, which is enabled by the proposed pretraining procedure.
>
>
> We sincerely appreciate your thorough review and the opportunity to clarify these points. Addressing these concerns will significantly improve the clarity and presentation of our work, ensuring a better understanding of our contributions. Please let us know if you have any further comments or suggestions.

---

### Official Review · Reviewer_TWiF · 2024-11-03

**Soundness:** 3
**Presentation:** 3
**Contribution:** 3
**Rating:** 6
**Confidence:** 3

**Summary:**

The paper introduce a novel routing policy that optimizes the selection of LLMs based on user-specific preferences. The authors frame the model selection problem as a multi-objective optimization task, balancing performance and cost. They propose a preference-conditioned routing mechanism that dynamically adapts to individual user preferences during inference.

The proposed method generalizes across various LLMs and user preferences, demonstrating significant improvements in both performance and cost-efficiency. For example, the method showed generalization to unseen model from HuggingFace OpenLLM
v2 benchmark.

**Strengths:**

1. Generalization Across Diverse Models and Tasks: One of the key strengths of the proposed routing mechanism is its ability to generalize effectively across a wide range of LLMs and various user queries. The authors leverage the model identity vector, which allows the routing policy to adapt to new and unseen LLMs seamlessly.

2. Model Identity Vector: The introduction of the model identity vector is another significant strength of the paper. This vector encapsulates the strengths and weaknesses of each LLM based on its performance across a diverse set of evaluation prompts. Which could be helpful in other tasks

**Weaknesses:**

While the reinforcement learning component lends credibility to the results, the generalization—particularly to unseen models—appears to be on the use of a model identity vector. This is a very interesting concept, as proposed in [1] as "model embedding". The authors could have strengthened their approach by comparing it to [1]. In that work, a systematic method is proposed for generate the identity vector, closely resembling the approach in this paper, which employs KL penalties alongside predictions for pairwise comparisons and binary evaluation outcomes.

[1] EmbedLLM: Learning Compact Representations of Large Language Models

**Questions:**

Is it possible to marry the identity vector with the method proposed in [1] to further improve the routing performance?

---

> ### Author Response · Authors · 2024-11-16
>
> Thank you for your insightful review and for acknowledging the strengths of our proposed preference-conditioned routing mechanism for LLMs, particularly its ability to generalize across diverse models, tasks, and user preferences. We appreciate you highlighting the significance of the model identity vector concept and its potential applicability to other tasks beyond routing.
>
> Regarding the weakness pointed out, we thank you for bringing the concurrent EmbedLLM work to our attention. We were not previously aware of this work, which does indeed propose a similar concept of learning compact representations (embeddings) for LLMs. In the revised version of our paper, we will include a detailed discussion of the EmbedLLM approach and compare our method for generating model identity vectors to their systematic framework.
>
> To your question about potentially marrying our identity vector approach with the EmbedLLM method, we find this to be an intriguing direction for future work. One possibility could be to use the EmbedLLM framework to learn a general-purpose embedding for each LLM, and then fine-tune these embeddings using our IRT-based approach to obtain task-specific identity vectors. Alternatively, we could explore incorporating the pairwise comparison data and KL regularization from our models into EmbedLLM training process to improve generalizability of the learned representations.
>
> We thank you again for bringing this related work to our attention and for the suggestion to explore potential synergies between the two approaches. Integrating ideas from EmbedLLM could further enhance the effectiveness and generalization capabilities of our routing mechanism.
>
> Please let us know if you have any other comments or suggestions. We appreciate your feedback and will incorporate it to improve our work.

---

### Official Review · Reviewer_1cFj · 2024-11-04

**Soundness:** 4
**Presentation:** 2
**Contribution:** 3
**Rating:** 6
**Confidence:** 4

**Summary:**

The paper introduces a cost-efficient routing framework for selecting Large Language Models (LLMs) by framing the selection process as a multi-armed bandit problem. This dynamic routing mechanism leverages user preferences to balance between cost and performance, adjusting selections based on individual needs and allowing for generalization to new, unseen models. The method uses a model identity vector to represent each LLM’s strengths across various tasks, and an efficient quizzing mechanism for integrating new models with minimal evaluation. Experimental results show the effectiveness of this approach in achieving optimal trade-offs in real-world applications​.

**Strengths:**

- This paper formulates a novel multi-objective routing mechanism that balances performance and cost through dynamic, preference-conditioned routing, capturing the Pareto front for user-specific trade-offs.

- This paper develops a flexible, action-space-aware routing policy that generalizes effectively across various LLM models, leveraging extensive evaluation data to handle diverse user queries and model configurations.

- This paper introduces an efficient quizzing method to rapidly integrate new models by assessing a small subset of prompts, demonstrating the routing mechanism’s superior adaptability and performance across tasks and domains.

**Weaknesses:**

- This paper proposes a sophisticated method for training the routing policy. While it is detailed, the complexity makes it challenging to track which data (prompts, embeddings, preference labels, target values, prediction scores, costs) is used for pretraining and training each component and how they are acquired. Although careful reading of different sections and the appendix clarifies these elements, it requires significant effort.

- The policy training does not explicitly address uncertainty in reward prediction, raising concerns that diverse prompts and a large sample space may be necessary to avoid suboptimal routing decisions.

**Questions:**

* Can the model identity vector be trained end-to-end with the preference-conditioned routing policy? The objectives of score prediction and model identity vector seem to overlap.

* The routing policy design might be innovative, but the remaining components feel traditional. Is there additional novelty in the proposed training algorithm or the data used for the routing policy?

* Can you elaborate more on how the policy can balance exploration and exploitation without explicitly modeling the uncertainty in the value functions?

---

> ### Author Response · Authors · 2024-11-16
>
> Thank you for your thorough review and insightful comments on our paper. We appreciate the acknowledgment of the novelty and significance of our proposed preference-conditioned dynamic routing mechanism for LLMs. We will strive to clarify the concerns raised and address the weaknesses identified in the revised version of the paper.
>
> Regarding the complexity of the training procedure, we agree that the method involves several components and data sources. To improve clarity, we will restructure the relevant sections and provide a more comprehensive overview of the training pipeline, explicitly detailing the data sources and their roles in pretraining and training each component. We will also consider adding a supplementary diagram or pseudocode to better illustrate the flow of data and components. Additionally, we plan to open-source our code to allow for transparent inspection of the data pipelines.
>
> Concerning the handling of uncertainty in reward prediction, we acknowledge that our current approach does not explicitly model this uncertainty. However, the prediction scores $p\hat{p}_k = \text{sigmoid}(f(x, I_k))$ obtained from the model identity vectors are provided as auxiliary information to the routing policy during training. By conditioning the policy on these prediction scores, it can implicitly learn to account for the uncertainty in the predictions. When the prediction scores are less confident (closer to 0.5), the policy will place less weight on them during routing decisions. Conversely, for prompts where the predictions are more certain, the policy can rely more heavily on the prediction scores. In this way, uncertainty is implicitly communicated to the policy through the prediction score values themselves.
>
> To address the question about joint training model identity vector and routing policy, we believe it can make training unstable. Since the model identity vectors effectively define the action space for the routing policy, allowing them to change during training makes the action space non-stationary, violating the Markov assumption and causing training to diverge. That said, we agree that end-to-end training could be an interesting future direction to explore more fully. Potential approaches could involve alternating optimization, where the identity vectors are kept fixed while training the policy, and then the vectors are updated while keeping the policy weights fixed. Techniques from the literature on non-stationary reinforcement learning may also provide insights.
>
> Regarding the novelty of the training algorithm and data used, we acknowledge that some components, such as the IRT model and the use of existing evaluation leaderboards, are adapted from prior work. However, we believe the primary novelty lies in the formulation of the preference-conditioned routing policy, the action-space-aware architecture, and the integration of diverse data sources (evaluation scores, pairwise comparisons, and user preferences) into a unified training pipeline. We will endeavor to better highlight these novel aspects in the revised version.
>
> Finally, concerning the exploration-exploitation balance, there are a couple of mechanisms at play. First, our routing policy outputs a distribution over the action space (selecting LLM $M_k$). During training, trajectories are sampled from this stochastic policy distribution, allowing for exploration of different actions. Second, the entropy regularization term in PPO explicitly encourages the policy to explore different actions. Even if the current value estimate $V_{\pi}$ for an action is low, there is a non-zero probability for the policy to still explore that action, which allows updating the value estimate if the actual reward contradicts the current estimate. Furthermore, by conditioning the policy on predicted scores $\hat{p}_k = f(x, I_k)$ from the model identity module, we provide an inexpensive signal about the expected reward for each action. This predictive information assists with guiding exploration towards more promising actions without requiring actual environment rollouts. Together, the stochastic policy, entropy regularization, and inexpensive predictive model scores enable the routing policy to inherently balance exploration of uncertain context-action pairs and exploitation of high-value estimates from past experience.
>
> We appreciate the reviewer's insightful comments and constructive feedback, which will undoubtedly improve the clarity and quality of our paper. We look forward to addressing the raised concerns in the revised version.

---

### Official Review · Reviewer_N8Q4 · 2024-11-04

**Soundness:** 3
**Presentation:** 3
**Contribution:** 3
**Rating:** 6
**Confidence:** 3

**Summary:**

This study introduces a novel framework for optimizing the selection of large language models (LLMs) based on user preferences, effectively balancing accuracy and cost. The research addresses the challenges associated with selecting appropriate LLMs, given their diverse capabilities and associated costs. By employing a multi-armed bandit approach with preference conditioning, the authors provide a flexible and efficient solution for optimizing LLM selection tailored to individual user needs. The results indicate significant improvements in both performance and cost management, positioning this framework as a promising avenue for future research in LLM utilization.

**Strengths:**

1. The proposed methods effectively address the issue of routing among multiple models (more than two), allowing for scalability beyond the RouteLLM baselines.
2. The authors present a pipeline consisting of model quizzing and preference-conditioned routing. For each component, they conduct optimization analysis and develop corresponding solution methods.
3. The comprehensive results across multiple tasks and domains demonstrate that the routing mechanism provides superior performance.

**Weaknesses:**

1. The experimental results presented solely in figures are difficult to interpret. The authors could enhance clarity by including results in tables along with relevant metrics.
2. The integration of a multi-armed bandit framework with dynamic routing based on user preferences introduces complexity to the overall system. This complexity may present challenges in terms of implementation, maintenance, and scalability, particularly for organizations with limited technical resources.
3. There is a lack of detailed exploration of the complex components of the routing policy, such as the specific parameter selection for model quizzing and comparison methods.

**Questions:**

1. The predictor baseline shows strong results when dealing with a family of LLMs in the experimental results. How should we interpret the differences between baselines in the "Family of LLMs" setting?
2. The additional computational requirements for dynamically generating and processing user preferences may lead to increased latency and resource consumption. Could the authors provide an efficiency analysis regarding this?
3. Data sensitivity is a critical issue. The effectiveness of dynamic routing heavily relies on accurately capturing user preferences. If users are unable to clearly articulate their needs or if their preferences change frequently, the system's performance may suffer due to misalignment between model selection and actual user requirements. How should datasets be prepared to address this?
4. RouteLLM provides MT-Bench results to demonstrate its effectiveness. According to the results in this study, RouteLLM appears to perform poorly on selected benchmarks. Could the authors provide a comparison on MT-Bench for both the baselines and the proposed methods?

---

> ### Author Response · Authors · 2024-11-16
>
> Thank you for your insightful review and feedback. We appreciate you raising these important points, and we will incorporate your suggestions into the revised manuscript to strengthen our work. Here are our responses to your specific comments:
>
> 1. Regarding the presentation of experimental results, we agree that including tabular results alongside the figures would enhance clarity and interpretability. In the revised manuscript, we will incorporate tables presenting relevant metrics for the different benchmarks and LLM configurations, complementing the existing figures.
> 2. We acknowledge your concern about the potential complexity introduced by integrating the multi-armed bandit framework with dynamic routing based on user preferences. To address this, we will provide an open-source implementation of our code, allowing others to examine and potentially extend our work. Additionally, we will include a discussion on the modular design and extensibility of our framework, highlighting how it can be efficiently implemented and scaled.
> 3. Concerning the exploration of the routing policy components, you make a fair point. We did not extensively tune hyperparameters for certain components, such as model quizzing and incorporating pairwise comparison data. The exact hyperparameters we used are provided in appendix. In the revised manuscript, we will acknowledge this limitation and emphasize that further improvements are possible through more comprehensive hyperparameter tuning.
> 4. Regarding the predictor baseline's performance, you raise an excellent point. In theory, if the performance prediction is accurate, the predictor policy should be optimal. Our routing policy aims to improve upon the predictor when those predictions are inaccurate or uncertain. For settings where our routing policy does not significantly outperform the predictor, it is likely due to the inherent difficulty in accurately predicting certain types of scores, such as discontinuous metrics like accuracy. We will include this analysis in the revised manuscript, as it provides valuable insights into the strengths and limitations of our approach.
> 5. Concerning the efficiency analysis, we appreciate you highlighting this aspect. Both the IRT model and routing policy are relatively lightweight, consisting of only a few linear layers. Therefore, the computational overhead introduced by our framework, compared to running LLM inference, is relatively low. We will quantify this overhead in the revised manuscript.
> 6. Regarding the MT-Bench comparison, thank you for bringing this up. We do have results on this benchmark, but we left them out of the initial submission as it is a relatively small dataset. Our routing policy obtains similar performance to RouteLLM. We agree that including these results would provide a more comprehensive evaluation. In the revised manuscript, we will add results on this dataset.
>
> We sincerely appreciate your thorough review and constructive feedback. Addressing these points will significantly strengthen our work and contribute to a better understanding of the proposed routing framework's capabilities, limitations, and potential areas for further improvement.
>
> Please let us know if you have any additional comments or suggestions.

---

> ### Comment · Reviewer_N8Q4 · 2024-11-30
>
> Thank you for the feedback. I decide to keep my score.

---

### Meta-Review · Area_Chair_XtJj · 2024-12-16

**Metareview:**

The paper proposes a bandit-based approach to routing amongst multiple LLMs for improved inference cost. The key idea is to first construct an identity vector for each LLM, based on a matrix factorization of performance on a set of pre-defined prompts. This identity vector is fed as input to a policy function that is trained via PPO on a set of evaluation leaderboards. The proposed method is amenable to generalization to new prompts and LLMs.

Reviewers were generally weakly supportive of the paper. The idea of constructing an LLM identity vector, and the resulting performance of the router, were seen as positives. One recurring concern was the complexity of the method, which involves several components. Another related concern was the technical explanations having scope for greater clarity, particularly around the precise reason the approach follows the bandit rather than supervised learning framework. The author response provided some clarifications, although there was a missed opportunity to incorporate changes into a revision for reviewers to consider.

From the AC's reading, we largely concur with the reviewers that the paper makes some interesting contributions, but has scope for greater clarity. We believe the findings in the paper could be useful for further research in this topical and important area, but that the impact could be greater with amendments. For example:
- the definition of a policy $\pi$ is imprecise, since it outputs a _distribution_ over $\{ 1, \ldots, K \}$
- the presentation of Section 2.3 prior to Section 2.4 may lead to some confusion in the present state, as it is not clear how $f( e_n, I_k)$ relates to $s( x_n, k )$
- in Equation 3 and elsewhere, the use of subscripts would enhance readability
- in Equations 2 and elsewhere, there is no specification of the argument being optimized over on the LHS
- the motivational claim in Section 2.4 that _"Unlike direct score prediction, this approach allows the policy to capture more nuanced information from the prompts and implicitly manage prediction uncertainty"_ is unclear
- introducing Equation 5 followed by Equation 6 is a little redundant, as the latter strictly subsumes the former
- by the end of Section 2.4, it is not clear what supervisory signals are provided to train the policy
- by the end of Section 2.5, it is not fully clear as to the distinction between the pre-training and training stages, and why one of them uses supervised learning and the other does not; this point is particularly important, as the introduction of a bandit framework is one of the paper's main stated contributions. The author response included the comment that _"the routing policy will choose one LLM to send the query"_, but this does not clarify why the supervision provided excludes the possibility of using supervised learning
- in Algorithm 1, mention is made of the evaluation dataset $\mathcal{D}$ comprising samples of the form $(x, \{ (M_k, c_k, s_k) \})$, but this is not easy to trace to any discussion in the text
- some of the citations could be expanded. For example, there are many other works on LLM routing and cascading; see, e.g.,

Gupta et al., Language model cascades: token-level uncertainty and beyond. ICLR 2024

Yue et al., Large Language Model Cascades with Mixture of Thoughts Representations for Cost-efficient Reasoning. ICLR 2024

Hu et al., RouterBench: A Benchmark for Multi-LLM Routing System. ICML Agentic Markets Workshop 2024

Overall, the paper is on the borderline. We believe the direction is extremely promising, and think there are some interesting ideas in the work. However, we think that the paper would benefit most from revisions that incorporate reviewer suggestions, which can undergo a fresh round of reviews.

**Additional Comments On Reviewer Discussion:**

During the discussion, the authors provided clarifications to several of the reviewer comments and questions. From the AC's reading, some of the clarifications around the substantive critiques, such as the precise need to employ bandit or RL algorithms rather than supervised algorithms, left some doubts. We also note that there was a missed opportunity to submit a revision that incorporated changes and clarifications.

---

### Decision · Program_Chairs · 2025-01-22

Reject